# OpenReview forum: "Omni-Reward: Towards Generalist Omni-Modal Reward Modeling with Free-Form Preferences"
_NeurIPS.cc/2025/Datasets_and_Benchmarks_Track — Submitted to NeurIPS 2025 Datasets and Benchmarks Track_

### Official Review · Reviewer_RQB9 · 2025-06-26

**Rating:** 4
**Confidence:** 3

**Summary:**

The paper proposes Omni-Reward to show the limitations of current RMs – a. limited support for modalities beyond image and text, such as video and audio and b. rigidity as preference is only capture by binary preference pairs. As part of this effort, they propose a. Omni-RewardBench across 5 modalities with free-form preferences b. Omni-RewardData, a large 248k sized preference training dataset and c. Omni-RewardModel, a collection of both discriminative and generative reward models.

**Additional Feedback:**

Overall, I find the paper interesting as it creates a diverse benchmark for reward modeling across 5 modalities and does a rigorous job on the techniques and models used for evaluating and training reward models. However, this is let down by substantial issues associated with human data collection in its current state.

**Dataset Code Accessibility:**

Yes

**Dataset Code Comments:**

Yes, links to data/code are readily available in the Abstract.

**Ethical Comments:**

1.	The license information for both existing assets that the authors built upon and for the newly created assets are not indicated. The README for all Huggingface/GitHub assets are also empty.
2.	Line 1151 also notes that an IRB was not obtained as data was collected from only internal annotators. Having worked on human preference data collection myself, a back of envelop calculation (assuming a minimum of 5 minutes per task) means that it at least takes 18625 minutes or over 300 hours of work per annotator (multiplied by 3 annotators). It’s not clear to me whether an IRB/ethical review is needed for this amount of work so raising it for an abundance of caution.
3.	For data quality and representativeness issue, see Weakness 3

**Ethical Considerations:**

Yes, there are significant ethics concerns that require review by an ethics expert

**Ethics Flags:**

["Improper research involving human subjects", "Data privacy, copyright, and consent", "Data quality and representativeness"]

**Final Justification:**

The rebuttal address some of my concerns detailed in the response to rebuttal.

There are a few outstanding issues

a. Claims of existing preference data to be overly rigid
b. Quality of GPT-4o annotations
c. Validity of w/Ties setup
d. Misleading column names on Huggingface Training Dataset

But these aspects are relatively minor and given my comments, hence recommend borderline accept in light of the pioneering nature of omni-modal benchmark and associated training data.

**Limitations Weaknesses:**

1.	The claim that existing preference data is rigid in Lines 4 and 41 seems a little unfair since there has been prior works that explore how to incorporate free-text criteria along binary preference labels such as [1] and [2]. It will be good to acknowledge these works and clarify how the paper differentiates from them.
2.	Upon looking at the preference data (https://huggingface.co/datasets/jinzhuoran/OmniRewardData/viewer/EvalMuse/train?row=1&views%5B%5D=evalmuse), it seems like both the “chosen” and “rejected” fields are identical across the first two samples. I’m not sure if this is a bug from the data upload or if this is an error during the dataset construction.
3.	The paper has limited information on how human annotation was carried out, making it difficult to get a sense of data quality. On line 61, the paper notes that “three annotators were employed to provide a free-form textual preference description and label the response as chosen, rejected or tied” to produce 3725 human-annotated pairs across 9 tasks and 5 modalities (text, image, video, audio and 3D). Line 178 further notes that the “entire annotation process is conducted by three PhD students in computer science”. The paper doesn’t report any chance-accounted measurement of inter-rater agreement (e.g. cohen’s/fleiss’ kappa or krippendorff’s alpha) other than removing 38% of samples with conflicting preferences. I find that such an approach towards quality control lacks rigor and could potentially introduce wrong labels into the test sets (when annotators agree by chance). There’s also no quality control on the preference criteria reported. My other concern is that all data is annotated by 3 CS PhD students and hence there might be a large extent of bias represented in this dataset due to the narrow set of annotator demographic.
4.	Looking more closely at the evaluation dataset https://huggingface.co/datasets/HongbangYuan/OmniRewardBench/viewer/text_audio_to_text/test?views%5B%5D=text_audio_to_text&row=4 , rows 4 and 5 have exactly the same prompt responses but opposite preferences (row4 prefers response2 while row5 prefers response1). This seems to be common (rows 6,7,8 also don’t have the same preference). The authors claim they remove “38% of samples with conflicting preferences” in Line 180 but the empirical dataset does not indicate this. This is also problematic for a benchmark as this means that it’s impossible for reward models to score 100% as the same input (prompts + response pairs) gives two opposite preference ground truths across two samples.
5.	The training dataset construction approach in Section 3.1 also doesn’t seem to ensure data quality. The approach first collates existing datasets, before augmenting the collection with GPT-4o-labelled-ground-truth preference pairs. They then verify it with 3 smaller models (4o-mini, Qwen2.5VL-7B and Gemma 3 12B), which might not enough as these models might not be as strong as GPT-4o and hence unable to catch mistakes made by GPT-4o. Such a curation approach also allows the model to reflect GPT-4o preferences, even when they are wrong (i.e. differ from human preferences).
6.	The paper does not explain how the w/Ties evaluation metric is calculated for BT models in Section 4.2, since BT models predict a scalar float score for each response at inference time and two responses very rarely get the same score. This means that BT models almost never predict “Ties”.

[1] Wang et al. (2025) HelpSteer2-Preference: Complementing Ratings with Preferences

[2] Liu et al. (2025) Inference-Time Scaling for Generalist Reward Modeling

**Strengths Contributions:**

1.	While there are many open-source benchmarks for text-only / text-image reward model evaluation, this is the first time that I have learned about a benchmark for evaluating RMs in audio and video. Such a benchmark would encourage the developments of RMs in these modalities.
2.	The collection of tasks within this benchmark is impressive, involving 9 different tasks with 5 modalities. The responses are also collected across 13 LLMs, 14 MLLMs and 27 text to image models among others. The authors also evaluate their benchmark across a large set of models.
3.	The training approach for Discriminative (BT) and Generative Reward models is reasonable.
4.	The performance of Omni-RewardModels on an external benchmark VL-RewardBench is good, becoming the new SOTA on it.

---

> ### Author Rebuttal · Authors · 2025-07-29
>
> We sincerely appreciate your positive assessment of our work, particularly your recognition that the paper is interesting as it creates a diverse benchmark for reward modeling across five modalities and does a rigorous job on the techniques and models used for evaluating and training reward models. At the same time, we sincerely and urgently request the opportunity to clarify some misunderstandings regarding the quality of our dataset.
>
> > The claim that existing preference data is rigid in Lines 4 and 41 seems a little unfair since there has been prior works that explore how to incorporate free-text criteria along binary preference labels such as \[1] and \[2]. It will be good to acknowledge these works and clarify how the paper differentiates from them.
>
> We appreciate your insightful comment and the references to these two works. **We truly appreciate the two works you mentioned. \[1] provides a solid and well-curated preference dataset, which serves as a strong foundation for reward model training. \[2] offers an innovative perspective on reward modeling by enabling models to generate evaluation criteria based on prompts.** We would like to take this opportunity to clarify both the similarities and key differences between our work and these earlier efforts.
>
> HelpSteer2-Preference \[1] adopts a set of predefined evaluation dimensions, such as helpfulness, correctness, coherence, complexity, and verbosity, which marks a meaningful step toward more nuanced preference modeling. However, this approach remains limited to a fixed set of criteria. In contrast, **Omni-Reward emphasizes free-form, user-specified textual criteria that are context-dependent and unconstrained, enabling more flexible and expressive modeling of human preferences**.
>
> GRM \[2] proposes an approach where the model automatically generates evaluation principles based on the prompt and then scores responses accordingly. This direction is conceptually aligned with our notion of free-form preferences. To the best of our knowledge, GRM’s training data has not been made publicly available. In contrast, **we openly release our evaluation benchmark, training dataset, and reward models**. Moreover, **while both of these works primarily focus on text-based tasks, our benchmark covers a broader range of modalities, including text, image, audio, and video, across nine diverse tasks**, offering a more comprehensive suite for evaluating multimodal reward models.
>
> \[1] Wang et al. HelpSteer2-Preference: Complementing Ratings with Preferences.
>
> \[2] Liu et al. Inference-Time Scaling for Generalist Reward Modeling.
>
> > Upon looking at the preference data, it seems like both the “chosen” and “rejected” fields are identical across the first two samples. I’m not sure if this is a bug from the data upload or if this is an error during the dataset construction.
>
> **We would like to clarify that the identical textual content in the chosen and rejected fields is intentional and not a dataset construction error**. EvalMuse is a text-to-image dataset, and we deliberately designed the chosen and rejected fields to share the same textual prompt. **This is because the distinction between the two samples lies not in the text but in the images field**.
>
> Each sample in the dataset contains two images: the first corresponds to the chosen image, and the second to the rejected image. This format allows us to maintain consistency across different tasks. By ensuring that both chosen and rejected fields contain textual content, we can process all samples in a unified manner during training, regardless of whether the differences stem from text or other modalities. We sincerely apologize for any misunderstanding this may have caused. **This format is intentionally adopted across all our multimodal generation tasks** **to ensure consistency and compatibility with our training framework**.
>
> > The paper has limited information on how human annotation was carried out, making it difficult to get a sense of data quality.
>
> We sincerely apologize for the confusion caused by the lack of clarity. As illustrated in Figure 4, our annotation pipeline consists of two key stages: Criteria Annotation and Preference Annotation. **Throughout these two stages, we removed a total of 38% of the samples to ensure data quality.**
>
> - In the **Criteria Annotation** stage, we filtered out **23%** of the samples that were annotated with criteria deemed either **too vague** or **overly specific**, as part of our **quality control on preference criteria**, since such criteria would undermine the overall consistency and utility of the preference data.
>
> - In the **Preference Annotation** stage, we further removed **15%** of the samples due to **disagreements among annotators**, where no consensus could be reached on the preferred output. **Following your suggestion regarding inter-rater agreement, we also report Krippendorff’s alpha, which reaches 0.701, indicating substantial agreement among annotators.**
>
> We acknowledge that the current annotation was conducted by a small group of PhD students. The annotation of the entire dataset was a time-intensive process that required substantial manual effort from our team. **Given the constraints of limited resources, we took extensive measures to ensure the quality and consistency of the annotations** in Appendix C.2, which helped significantly mitigate potential biases. We will incorporate additional details on data quality based on your feedback.
>
> &#x20;
>
> > Looking more closely at the evaluation dataset, rows 4 and 5 have exactly the same prompt responses but opposite preferences (row4 prefers response2 while row5 prefers response1). This seems to be common (rows 6,7,8 also don’t have the same preference).
>
> We would like to clarify that this is not a construction error in the dataset. **While the prompts and response pairs may appear identical across certain samples, the inputs are not exactly the same because they differ in the criteria field, which reflects varying preference dimensions**. **This design is intentional and aligns with our goal of modeling free-form, criterion-dependent human preferences.** In other words, even with the same prompt and response pair, different evaluation criteria can lead to opposite preference labels\*\*.&#x20;
>
> > The training dataset construction approach in Section 3.1 also doesn’t seem to ensure data quality.
>
> We agree with your concern. To construct a large-scale training set, we employed GPT-4o to generate free-form instructions _I_ that reflect a user preference supporting either _y1_ or _y2_, along with the corresponding label _p_. **To mitigate potential systematic biases introduced by relying solely on GPT-4o, we incorporated a multi-model verification process** to mitigate potential errors and biases introduced by GPT-4o during instruction generation. Notably, **this filtering process is framed as a classification task, which is generally less complex and more robust than open-ended instruction generation, helping catch mistakes made by GPT-4o**.
>
> Moreover, generating preference data synthetically is a **common and practical approach** widely used in recent alignment and reward modeling research \[2, 3, 4, 5], especially when large-scale human annotation is infeasible. In addition, **the reward model trained on Omni-RewardData achieves strong performance not only on Omni-RewardBench, but also on other widely used reward model benchmarks, which further demonstrates the effectiveness and generalizability of our synthetic data**.
>
> \[3] Liu et al. Skywork-Reward: Bag of Tricks for Reward Modeling in LLMs.
>
> \[4] Yu et al. RLAIF-V: Open-Source AI Feedback Leads to Super GPT-4V Trustworthiness.
>
> \[5] Ji et al. Align Anything: Training All-Modality Models to Follow Instructions with Language Feedback.
>
> > The paper does not explain how the w/Ties evaluation metric is calculated for BT models in Section 4.2, since BT models predict a scalar float score for each response at inference time and two responses very rarely get the same score. This means that BT models almost never predict “Ties”.
>
> This is an insightful question. Following prior work, we do not rely on exact matching of scalar float scores for BT models. Instead, we adopt Algorithm 1 proposed in the Ties Matter paper \[6]. **This method systematically traverses a range of tie thresholds, computes the corresponding three-class accuracy (win/loss/tie) for each threshold, and selects the maximum accuracy as the final metric.** This approach allows for meaningful evaluation of ties, even when exact float equality is rare.
>
> \[6] Deutsch et al. Ties Matter: Meta-Evaluating Modern Metrics with Pairwise Accuracy and Tie Calibration.
>
> > The license information for both existing assets that the authors built upon and for the newly created assets are not indicated. The README for all Huggingface/GitHub assets are also empty.
>
> Thank you for pointing this out. We took this matter seriously and have now completed the necessary updates. **License information and documentation have been added to all relevant HuggingFace and GitHub assets to ensure transparency and compliance.**
>
> > Line 1151 also notes that an IRB was not obtained as data was collected from only internal annotators. Having worked on human preference data collection myself, a back of envelop calculation (assuming a minimum of 5 minutes per task) means that it at least takes 18625 minutes or over 300 hours of work per annotator (multiplied by 3 annotators).
>
> Thank you for your thoughtful and cautious remark. **We confirm that all annotations were conducted voluntarily by the authors of this paper, who were fully informed about the nature and purpose of the task, their rights, and how the data would be used**. We also followed the standard research ethics protocols of our institution for internal annotation efforts. We will clarify this in the final version of the paper to avoid any confusion.

---

> > ### Comment · Reviewer_RQB9 · 2025-08-02
> > **Response to Rebuttal**
> >
> > 1. The characterization of HelpSteer2-Preference is plain wrong, please refer to Appendix D to see that there is actually a free text field describing the reason for preference. An example is attached below. The main point I making here is that it's not right to claim to "existing preference data is rigid". It's ok to say something like while some explorations about alternative flexible representations of preference have been made, often they do not have openly released data or are limited to text-only data. This makes the contribution of the paper clearer without overclaiming.
> >
> > ```
> > {
> >             "statement": "@Response 2 is better than @Response 1 because it provides a comprehensive insightful explanation of signanal and its properties.",
> >             "elaboration": "It is complete, clear and correct as it discuss all the the poperties of signal while @Response 1 only discusses  three properties of signal. It does not diuscuss important properties like noise, phase and envelope. @Response 2  follows all the instruction but @Response 1 does not follow all the instruction. For instance the instruction requires an explanation of signal and its properties with an aid of a diagram but @Response 1 does not provide the diagram.",
> >             "strength": 1
> > }
> > ```
> >
> > 2. I checked the data again and while I understand the rationale better now, having it in this format is still confusing to people when they first look at the data. If the difference lies in the images, then one image should be called chosen and the other should be called rejected. The goal of releasing a dataset on huggingface is not only (and in my opinion, mainly) for the data to work well with your codebase. Instead, it's for people using various codebases to have a clear understanding of the data so that they can use it.  To achieve the first objective with your codebase, it's trivial to write some transformation logic within your code that does this conversion.
> >
> > 3. Thanks for clarifying on human annotations - I now have more confidence in the quality of the human annotations.
> >
> > 4. I'm still unconvinced that much smaller VLM (<12B) can catch errors that GPT-4o makes. It might be better to use larger, more powerful VLM to verify a sub-sample of it. Even better would be to ask a group of humans to verify it so that to understand the degree of noise contained within the GPT-4o annotations. While I understand many other datasets are build on synthetic data, this remains a limitation as the training data will be biased from it -and in many cases - if the benchmark data is also from synthetic using a similar model, it can reflect the bias without being a better VLM. The bias of using GPT-4o should also be noted in the limitations section.
> >
> > 5. Re. Ties, in more than 10 benchmarks on Reward Models that I've seen, I haven't seen any that adopts this w/ Ties approach. The problem is that ties and A is slightly better than B is a very narrow gap and in these situation, it excludes a lot of models that may not necessarily be able to predict ties. I would recommend focussing on the W/o Ties Benchmark solely. The threshold for different types of models are also hard to establish robustly between BT models and GenRM.
> >
> > 6 and 7. Thanks for clarifying the additional information.
> >
> > Given the discussion above, I will increase my rating accordingly. Since there are dedicated Ethics Reviewers for this submission, I will not further comment on those areas. What I hope to remind the authors (for future submissions) is to check that the submission is complete including the dataset/code README/licensing at submission time (and not rebuttal time). It's unclear to me if/how the track chair will penalize the paper for this slip this time but doing this can be risky in the future.

---

> > > ### Author Response · Authors · 2025-08-04
> > > **Response to Reviewer RQB9**
> > >
> > > Dear Reviewer RQB9,
> > >
> > > Thank you so much for your careful and insightful feedback and for raising the score! We are glad that our responses addressed your concerns. Below is our follow-up response.
> > >
> > > 1.  We sincerely apologize for the inaccurate description of HelpSteer2-Preference. Upon revisiting the paper and Appendix D, we fully acknowledge that it includes a free-text field for annotators to explain their preferences, which is both forward-thinking and practically valuable. Thank you for pointing this out. We will revise our description accordingly to better reflect the contributions of this work and more accurately position our own.
> > >
> > > 2.  We will update the dataset accordingly to make it easier for users to understand the data structure.
> > >
> > > 3.  We will follow your suggestion by adding an analysis of the consistency between GPT-4o-generated annotations and human evaluations. This will help better understand the potential noise in the training data generated by GPT-4o. We will also explicitly acknowledge the possible bias introduced by relying on GPT-4o in the Limitations section.
> > >
> > > 4.  We agree with your point that the use of ties is indeed uncommon in reward model evaluations for text generation, and is more frequently seen in image or video generation tasks, where subjective ambiguity is more prevalent. In the revised version, we will focus more explicitly on the results under the w/o Ties setting. We sincerely appreciate your thoughtful suggestion.
> > >
> > > Your advice is inspiring and has provided us with guidance towards a more comprehensive paper. We truly appreciate your support for our work.
> > >
> > > Best regards,
> > >
> > > Authors

---

### Official Review · Reviewer_HrXc · 2025-07-01

**Rating:** 4
**Confidence:** 2

**Summary:**

This paper introduces Omni-Reward, a detailed framework for omni-modal reward modeling that addresses two fundamental challenges in current reward models: modality imbalance (limited support for video, audio, and 3D modalities) and preference rigidity (inability to adapt to diverse user preferences). The work consists of three main components: (1) Omni-RewardBench - a benchmark with 3,725 preference pairs across 9 tasks and 5 modalities (text, image, video, audio, 3D) with free-form preference annotations, (2) Omni-RewardData - a multimodal preference dataset with 248K general preference pairs and 69K instruction-tuning pairs, and (3) Omni-RewardModel - both discriminative and generative reward models trained on the constructed dataset.

**Dataset Code Accessibility:**

Yes

**Dataset Code Comments:**

The dataset can be accessed through the hugging face

**Ethical Comments:**

As the authors states in the Appendix B, some preference pairs in Omni-Reward may contain offensive, inappropriate, or otherwise sensitive prompts and responses, as they are intended to reflect real-world scenarios.

**Ethical Considerations:**

No, there are no or only very minor ethics concerns

**Final Justification:**

The authors' rebuttal address most of my concern, thus I keep my initial score.

**Limitations Weaknesses:**

1. The benchmark size could be one problem. 3725 preference pairs may be insufficient for training large-scale models or supporting evaluations requiring millions of examples.
2. Heavy reliance on GPT-4o for generating instruction-tuning data may introduce systematic biases into the training set.
3. Uneven distribution of preference pairs across modalities may lead to biased model performance, as showing in Table 5 that some modalities have almost twice as many examples as others (529 vs 229 pairs).

**Strengths Contributions:**

1. The paper is overall well-written and easy to follow.
2. The  benchmark covering 9 distinct tasks across 5 modalities, significantly expanding beyond existing text and image-focused benchmarks.
3. The paper provides a rigorous evaluation framework that reveals substantial performance gaps in current models, particularly for underexplored modalities like audio and 3D.
4. The evaluation is comprehensive which contains 30+ models including state-of-the-art commercial models (GPT-4o, Claude 3.5 Sonnet) and specialized reward models.

---

> ### Author Rebuttal · Authors · 2025-07-28
>
> Thank you for your careful and insightful reviews. Your comments have provided valuable inspiration and direction for improving our work. We sincerely hope that our response addresses your concerns.
>
> > The benchmark size could be one problem. 3725 preference pairs may be insufficient for training large-scale models or supporting evaluations requiring millions of examples.
>
> Thank you for your comment. We would like to emphasize that the **entire benchmark is fully human-annotated**, guided by carefully defined criteria and consistent annotation protocols. The annotation process was rigorous and time-consuming, requiring substantial manual effort from our team to ensure the quality, consistency, and reliability of the data. Unlike automatically generated preference data, our benchmark captures rich, fine-grained human preferences across multiple dimensions. As shown in Table 4, the size of our benchmark is comparable to other human-annotated reward model datasets, indicating that it aligns with the standard scale for high-quality, manually curated evaluation benchmarks.
>
> &#x20;
>
> **It is also important to note that our benchmark is intended primarily for evaluation, rather than for large-scale training**. To support large-scale training of reward models, we construct Omni-RewardData, a complementary multimodal preference dataset comprising 317K preference pairs. By training reward models at scale using this dataset, we further demonstrate the scalability and applicability of our data, as well as its effectiveness in improving model performance across diverse modalities and tasks. **Together, this combination of a carefully curated human evaluation benchmark and a large-scale training dataset enables both rigorous assessment and practical development of reward models in omni-modal settings.**
>
> > Heavy reliance on GPT-4o for generating instruction-tuning data may introduce systematic biases into the training set.
>
> We agree with your concern. To construct a large-scale training set, we employed GPT-4o to generate free-form instructions _I_ that reflect a user preference supporting either _y1_ or _y2_, along with the corresponding label _p_. **To mitigate potential systematic biases introduced by relying solely on GPT-4o, we incorporated a multi-model verification process.** Specifically, we used GPT-4o-mini, Qwen2.5-VL-7B, and Gemma-3-12B-it to verify the consistency of each tuple _(I, x, y1, y2)_ with its label _p_, in order to mitigate potential errors and biases introduced by GPT-4o during instruction generation.
>
> Moreover, generating preference data synthetically is a **common and practical approach** widely used in recent alignment and reward modeling research \[1, 2, 3], especially when large-scale human annotation is infeasible. In addition, **the reward model trained on Omni-RewardData achieves strong performance not only on Omni-RewardBench, but also on other widely used reward model benchmarks, which further demonstrates the effectiveness and generalizability of our synthetic data**.
>
> &#x20;
>
> \[1] Chris Yuhao Liu, Liang Zeng, Jiacai Liu, Rui Yan, Jujie He, Chaojie Wang, Shuicheng Yan, Yang Liu, Yahui Zhou. Skywork-Reward: Bag of Tricks for Reward Modeling in LLMs.
>
> \[2] Tianyu Yu, Haoye Zhang, Qiming Li, Qixin Xu, Yuan Yao, Da Chen, Xiaoman Lu, Ganqu Cui, Yunkai Dang, Taiwen He, Xiaocheng Feng, Jun Song, Bo Zheng, Zhiyuan Liu, Tat-Seng Chua, Maosong Sun. RLAIF-V: Open-Source AI Feedback Leads to Super GPT-4V Trustworthiness.
>
> \[3] Jiaming Ji, Jiayi Zhou, Hantao Lou, Boyuan Chen, Donghai Hong, Xuyao Wang, Wenqi Chen, Kaile Wang, Rui Pan, Jiahao Li, Mohan Wang, Josef Dai, Tianyi Qiu, Hua Xu, Dong Li, Weipeng Chen, Jun Song, Bo Zheng, Yaodong Yang. Align Anything: Training All-Modality Models to Follow Instructions with Language Feedback.
>
> > Uneven distribution of preference pairs across modalities may lead to biased model performance, as shown in Table 5 that some modalities have almost twice as many examples as others (529 vs 229 pairs).
>
> Thank you for pointing this out. Due to the inherent difficulty of collecting high-quality data across multiple modalities, some imbalance in the distribution of preference pairs is unavoidable. **While some imbalance remains, our dataset maintains a relatively balanced distribution across modalities, especially when compared to the significant disparities commonly observed in real-world data availability between modalities such as images and audio**. Moreover, in our evaluation, we report **per-task performance** to ensure that models are assessed fairly within each modality, thereby reducing the impact of sample size differences on overall performance comparisons. At the same time, we will also follow your suggestion and expand the dataset to achieve a more balanced distribution across modalities as much as possible.

---

> > ### Comment · Area_Chair_n2gd · 2025-08-05
> >
> > Dear Reviewer,
> >
> > Please engage in the rebuttal discussion and indicate whether the authors' response addresses your concerns. Thank you for your time and effort.
> >
> > Best,
> > Area Chair

---

> > ### Comment · Reviewer_HrXc · 2025-08-06
> >
> > I appreciate the authors' responses on my comments. The responses addressed my concern, thus I keep the positive score as initial.

---

> > > ### Author Response · Authors · 2025-08-06
> > > **Appreciate your new response**
> > >
> > > Dear Reviewer,
> > >
> > > Thank you so much for your careful and insightful feedback! We are glad that our responses addressed your concerns. Your advice is inspiring and has provided us with guidance towards a more comprehensive paper. We truly appreciate your positive score for our work.

---

### Official Review · Reviewer_8Bok · 2025-07-02

**Ethics Flags:** Improper research involving human sub…
**Rating:** 5
**Confidence:** 3

**Summary:**

The submission contributes a new dataset, benchmark, and model for predicting preferences. The dataset and benchmark are focused on any-to-any modality models and annotations that can be contextually guided to focus on specific preference dimensions. The new model is trained on this new dataset and performs well relative to a wide range of open source and commercial models.

**Additional Feedback:**

I am putting heavy weight on the quality of the code / data release and engagement on prior work as I recommend rejection of the paper in the current state -- but I think both of these are solvable with a modest amount of effort, so I am optimistic that the authors can address them and am very open to increasing my score by multiple points if these issues are improved.

EDITED: improved, thanks.

**Dataset Code Accessibility:**

Yes

**Dataset Code Comments:**

The code / data / model releases have no readmes, model cards, usage instructions, etc. For a datasets and benchmarks paper especially, these should be detailed and high quality. At minimum, the release should provide clear licenses on the code and data, readme containing data dictionaries (what do the fields mean?) and usage examples (how do I train on the data? How do I use the benchmark?).

EDITED: this is much improved, thanks.

**Ethical Comments:**

Per NeurIPS ethics guidelines, research involving human participants (such as annotation) should follow standard research ethics protocols in their institution (e.g. human subject research accreditation or institutional review board), or otherwise an equivalent informal process. The submission should include some statement to the effect that such a process was followed (this can be in the supplement).

**Ethical Considerations:**

Yes, there are ethics concerns that require attention by the authors

**Final Justification:**

I appreciate that the authors addressed a majority of my suggestions both related to data quality / availability, and experiments showing the benefits of OmniRewardBench. I do have some remaining concerns given the size of the annotator pool, but as long as this is not hidden in the paper, the community can decide if this is a sufficiently representative benchmark and / or build upon in.

I remain slightly concerned about the ethics points I raised but defer to the ethics reviewer and AC on whether the authors' responses are sufficient.

**Limitations Weaknesses:**

To me the biggest weakness in the contribution is relatively minimal engagement with the AlignAnything paper of Ji et al (2024) and its EvalAnything benchmark. Like the present contribution, that work is concerned with any-to-any alignment. Also like the present contribution, it provides a new dataset and benchmark. Finally, like the present contribution, it rates generations along multiple dimensions (though those dimensions themselves are fixed and not contextually varying). I think it's plausible that the submission is both novel and useful relative to AlignAnything and EvalAnything, but further discussion and/or analysis is needed -- currently that work is backgrounded in one reference in the main text, plus a table in the supplement. For example, some analysis could be provided showing that the hybrid annotation of AlignAnything is inferior to the fully human annotation of the submission; or the submission could demonstrate that the new benchmarks ranks SOTA models differently, and provide some evidence or insight that the free-form preferences enable this. Or perhaps it could show quantitatively (rather than just with examples) that Omni-RewardBench contains many examples where the freeform preference reverses the choices, that these examples in particular are where models tend to fail, and that therefore this part of the contribution is important. More generally, having some information on how many different preference contexts there are per pair of generations would be useful in the paper somewhere. Furthermore, I think EvalAnything seems to be a better benchmark for Omni-RewardModel than VL-RewardBench due to the obvious similarities to the current work (though I think this is not critical, I see the new dataset and benchmark as core and the new reward model as a demonstration that they are useful).

Another related work that is briefly cited but not discussed is the RPR paper from Pitis et al. (NeurIPS 2024). There, as in the submission, preferences are contextual (or "freeform" as the submission describes them). RPR is text-only but should probably be mentioned in table 4 and discussed in context of preference rigidity.

I have some additional minor points that may help improve the work:
* Why use MiniCPM as base? It seems like Qwen2.5-Omni is a similar size and performs better. If the new dataset provides a similar lift to that model as it does to MiniCPM, the resultant model may be fully SOTA. Currently the model is competitive and SOTA in some mappings but far from all.
* I'm not sure that "Omni-modal" is an accepted term yet (though it seems growing in popularity). Perhaps it is good to clarify that it means any-to-any modality mapping.
* The paper notes that almost 40% of samples are removed due to rater disagreement. This seems very high to me. It may be interesting to release and analyze these examples rather than discard them, because it's a significant proportion and may provide useful insights about the nature of contextual "freeform preferences" that are generated
* $c$ is used for user preference in section 2.1 but then $I$ is used starting in section 3.1. Are these the same or different? If they are different, it would be useful to make this clear.
* A spider chart might be good in addition to Table 1 (maybe selecting a subset of top models). This would help support the claims starting line 271 about modality imbalance across tasks.
* How is overall score computed when there are missing modalities? It's not clear to me how this is handled, and makes the overall score hard to use to rank models. For example, Claude 3.5 is bolded as SOTA overall but GPT-4o beats it on 6 of the 8 modalities they share (and then performs worse on the additional modalities Claude 3.5 does not support).
* For the ablation, the statement in the text and table 3 make it a little hard to understand what's going on, since it appears that there are two models in the ablation (MiniCPM and Omni-RewardModel). But in fact, MiniCPM is the base, and then there are 5 different data mixes in the ablation (2 unimodal, 1 multimodal, preference-only, and full). Perhaps the text and especially table can be clarified along these lines.
* What data went into Figure 2? Is it data from table 1, i.e. this is correlation among models? This should be clarified.
* Fig 4 "Resposne" -> Response
* The authors might want to spot check the examples in the supplement. For example, Fig 16 talks about owlfolk in the prompt, not an instruction about coffee / beer. I'm also not sure if the annotation on Fig 13 is correct?

**Strengths Contributions:**

The submission is generally comprehensive and detailed: as befits an any-to-any modality focus, the dataset is assembled is filtered across a variety of different benchmarks, and the number of models considered is likewise large. While the performance of the new model isn't strictly SOTA across all modalities, it is competitive and illustrates both the sensitivity of the benchmark and the quality of the new dataset. Performance on VL-RewardBench is likewise strong, and some ablations are provided showing the strength of the design choices made w.r.t. the dataset. The appendix is likewise detailed with annotation prompts and dataset examples.

---

> ### Author Rebuttal · Authors · 2025-07-30
>
> Thank you for your careful and insightful reviews. We truly appreciate the valuable, detailed, and constructive suggestions you provided. They reflect a high level of expertise and thoughtful engagement with our work. We sincerely hope that our response addresses your concerns.
>
> > To me the biggest weakness in the contribution is relatively minimal engagement with the AlignAnything paper of Ji et al (2024) and its EvalAnything benchmark.
>
> We sincerely apologize for not engaging in a more in-depth discussion of AlignAnything due to space limitations. **We fully acknowledge that AlignAnything is a highly influential and insightful work, and it has indeed inspired several key aspects of our Omni-Reward contribution, particularly the concept of any-to-any alignment**. First, we would like to acknowledge several important **similarities** between Omni-Reward and AlignAnything:
>
> 1.  Both works aim to tackle the challenge of any-to-any modality alignment.
> 2.  Both works offer a comprehensive suite, including evaluation, dataset and model.
> 3.  Both works perform multi-dimensional evaluation, assigning scores along several aspects of generation quality.
>
> Secondly, we highlight the key **difference** between our work and AlignAnything. **We truly appreciate the insightful points you raised, they have provided valuable inspiration for us**. In addition to those perspectives, we observe that **AlignAnything primarily focuses on aligning all-modality models, aiming to improve their general capability across diverse input and output modalities**. In contrast, **our work places greater emphasis on the reward modeling process within the alignment pipeline**. For instance, AlignAnything introduces EvalAnything to evaluate the performance of aligned models across modalities. In comparison, OmniRewardBench is specifically designed to evaluate reward models, assessing whether their preferences are aligned with real-world human judgments under specific textual criteria.
>
> > For example, some analysis could be provided showing that the hybrid annotation of AlignAnything is inferior to the fully human annotation of the submission
>
> AlignAnything utilizes a hybrid annotation approach, designed for generating large-scale training datasets. In contrast, our method aims to develop a higher-quality benchmark that aligns with human preferences for evaluating the reward models. Therefore, **our benchmark requires higher data annotation standards, necessitating human involvement in the annotation process**.
>
> > The submission could demonstrate that the new benchmarks ranks SOTA models differently, and provide some evidence or insight that the free-form preferences enable this.
>
> This is an excellent suggestion. **We compare the performance of ten models on OmniRewardBench and VLRewardBench, obtaining a Spearman correlation coefficient of 0.4572 between their rankings**. This indicates that incorporating additional modalities and free-form criteria differentiates our benchmark from previous ones.
>
> > Or perhaps it could show quantitatively (rather than just with examples) that Omni-RewardBench contains many examples where the freeform preference reverses the choices, that these examples in particular are where models tend to fail, and that therefore this part of the contribution is important.
>
> You raised a very important point that touches on the core of our contribution. To demonstrate the challenge posed by our benchmark, **we present a quantitative experiment showing that, despite the use of free-form criteria, the model tends to rely more on its inherent judgment than on the provided criteria**. Specifically, we first collect the model’s inherent preferences without any criteria. Based on their alignment with the ground-truth annotations, we divide the samples with criteria into two groups: those where the inherent and criteria-based preferences are consistent, and those where they conflict. We then evaluate the model’s accuracy under the criteria for both groups and observe a significantly lower performance in the inconsistent group. **The experimental results demonstrate that for GPT-4o-mini, the accuracy on inconsistent samples is lower than that on consistent samples by 26.32% on average, and for Claude-3-5-Sonnet, it is 18.50%**.
>
> > More generally, having some information on how many different preference contexts there are per pair of generations would be useful in the paper somewhere.
>
> For each preference pair in every task, the average number of preferences is **3.57**. In future versions, we will provide a more detailed statistical table.
>
> > EvalAnything is a more suitable benchmark for Omni-RewardModel than VL-RewardBench.
>
> We agree that EvalAnything is a more comprehensive and suitable benchmark for evaluating omni-modal models. Our original choice of VL-RewardBench was based on the fact that VL-RewardBench is specifically designed to assess the performance of reward models, whereas EvalAnything is primarily intended to evaluate aligned models that handle any-to-any multimodal tasks. We plan to incorporate EvalAnything in our evaluation.
>
> > RPR also uses contextual preferences and should be included in Table 4 and discussed in relation to preference rigidity.
>
> Thanks for your valuable advice. **We should certainly add the RPR work to the comparison table to provide a more comprehensive comparison**.
>
> > Why use MiniCPM as base?
>
> Our training work began relatively early, and at that time, we selected MiniCPM-o-2.6 as the base model due to its balanced size and strong performance. We fully agree with your observation that Qwen2.5-Omni-7B may be a stronger base model. Following your suggestion, we have conducted additional training using Qwen2.5-Omni-7B as the base model. **The experimental results show that Qwen2.5-Omni-7B achieves a 3.4% improvement in overall performance on Omni-RewardBench, compared to MiniCPM-o-2.6**.
>
> > I'm not sure that "Omni-modal" is an accepted term yet.
>
> We agree that the term "omni-modal" may not yet be formally defined, even though it is gaining popularity in recent literature. To ensure clarity, we provided an explanation in Line 21, defining it as an any-to-any models. **We fully agree that the definition of any-to-any modality introduced in AlignAnything provides a clearer and more precise understanding of the concept**.
>
> > Nearly 40% of samples were removed due to rater disagreement, which is unusually high and suggests these samples should be released and analyzed for insights into contextual freeform preferences.
>
> As illustrated in Figure 4, our annotation pipeline consists of two key stages: Criteria Annotation and Preference Annotation. **Throughout these two stages, we removed a total of 38% of the samples to ensure data quality.**
>
> - In the **Criteria Annotation** stage, we filtered out **23%** of the samples that were annotated with criteria deemed either **too vague** or **overly specific**, as part of our **quality control on preference criteria**, since such criteria would undermine the overall consistency and utility of the preference data.
>
> - In the **Preference Annotation** stage, we further removed **15%** of the samples due to **disagreements among annotators**, where no consensus could be reached on the preferred output.
>
> To support future research, we plan to release these excluded samples along with disagreement annotations in a supplementary dataset.
>
> > $c$ is used for user preference in section 2.1 but then $I$ is used starting in section 3.1. Are these the same or different? If they are different, it would be useful to make this clear.
>
> Thank you for pointing this out. In fact, $c$ and $I$ refer to the same underlying concept of user preference. We use $c$ in the context of evaluation data to denote human-annotated preferences, while $I$ is used in the training data to represent model-generated preference instructions. We will clarify this distinction in the revised version to avoid confusion.
>
> > A spider chart might be good in addition to Table 1 (maybe selecting a subset of top models).
>
> We agree that this type of visualization could effectively highlight modality-level differences, and we will incorporate it in future revisions.
>
> > The method for computing overall scores with missing modalities is unclear.
>
> **Our current approach to computing the overall score is to take the average performance across the modalities that a given model supports**. We acknowledge that this may not be an optimal strategy, and it can indeed lead to the issue you pointed out. We also experimented with assigning a score of zero to missing modalities, but this approach appeared unfair to models that simply do not support certain modalities (e.g., audio). Another possible solution we are considering is to report and compare omni-modal models and vision-language models separately. **We would be very interested in hearing any suggestions you may have for a more principled way to handle this challenge**.
>
> > For the ablation, the statement in the text and table 3 make it a little hard to understand what's going on, since it appears that there are two models in the ablation (MiniCPM and Omni-RewardModel).
>
> You are absolutely right that MiniCPM serves as the base model for Omni-RewardModel, and the remaining ablation experiments are conducted by training MiniCPM on different data mixtures.
>
> > What data went into Figure 2? Is it data from table 1, i.e. this is correlation among models?
>
> Yes, the data in Figure 2 is derived from Table 1, and it reflects the correlation of performance across models.
> For TA2T and T2A, we compute the correlation using only the models that support audio modalities. For the other tasks, we include all models in the correlation analysis to capture broader trends across modalities.
>
> > The code / data / model releases have no readmes, model cards, usage instructions.
>
> Thank you for pointing this out. **We took this matter seriously and have now completed the necessary updates**.

---

> ### Author Response · Authors · 2025-08-05
>
> Dear Reviewer 8Bok,
>
> We sincerely appreciate your thoughtful and constructive feedback. Your suggestions have been extremely valuable to us, and we have learned a great deal through the rebuttal process.
> Due to space constraints in the rebuttal, we had to keep some of our responses concise. However, we would be more than happy to provide additional details or clarifications if needed.
> We truly value your perspective and look forward to your continued consideration of our submission.
>
> Best regards,
>
> Authors

---

> ### Comment · Area_Chair_n2gd · 2025-08-05
>
> Dear Reviewer,
>
> Please engage in the rebuttal discussion and indicate whether the authors' response addresses your concerns. Thank you for your time and effort.
>
> Best,
> Area Chair

---

> ### Author Response · Authors · 2025-08-07
> **Looking Forward to Your Reply**
>
> **Dear Reviewer 8Bok,**
>
> Thank you very much for your insightful and constructive feedback. We truly appreciate your expertise in alignment, and we have learned a great deal from your comments and suggestions.
>
> To further improve our work, we would be grateful to know whether our responses have sufficiently addressed your concerns. We sincerely welcome any further discussion or suggestions you may have.
>
> Below is a brief summary of the key revisions we have made based on your feedback:
>
> 1. We have provided a detailed discussion comparing **Omni-Reward** and **AlignAnything**, including their similarities and respective limitations. We also added a discussion of the relationship between Omni-Reward and **RPR**.
> 2. We clarified the descriptions of our *code*, **benchmark**, **dataset**, and **model**, and provided clear **usage examples** to facilitate understanding and adoption.
> 3. We further verified the intended usage of **AlignAnything**, and clarified that **EvalAnything** is primarily designed for evaluating aligned models capable of handling **any-to-any multimodal tasks**.
> 4. Following your suggestion, we retrained our reward model using **Qwen2.5-Omni-7B** as the base model and demonstrated its effectiveness through updated results.
> 5. We improved the clarity and precision of several technical details throughout the paper, thanks to your careful suggestions.
>
> We truly appreciate your valuable insights, and thank you again for your time and effort in reviewing our work.
>
> Best regards,
> The Authors

---

> ### Author Response · Authors · 2025-08-08
> **Confirmation on Addressing Your Concerns**
>
> Dear Reviewer 8Bok,
>
> Thank you again for your time and review.
>
> We noticed that you have clicked the “Mandatory Acknowledgement” button. We would like to kindly ask whether the additional experiments and clarifications provided in our rebuttal have sufficiently addressed your concerns. If you have any remaining questions, we would be very happy to engage in further discussion.
>
> Best regards,
>
> Authors

---

### Official Review · Reviewer_FwbR · 2025-07-19

**Rating:** 4
**Confidence:** 4

**Summary:**

The paper introduces Omni-Reward, a comprehensive framework aimed at advancing reward models (RMs) for aligning AI systems. It contains 9 tasks across 5 modalities (text, image, video, audio, 3D) and uniquely incorporates free-form preferences, where the evaluation criteria are specified in natural language.

**Dataset Code Accessibility:**

Yes

**Ethical Considerations:**

No, there are no or only very minor ethics concerns

**Limitations Weaknesses:**

1. Some limitations are listed in Appendix A. They are all valuable future directions. Besides, the work could benefit from including other important modalities like thermal, radar, tabular data, or time-series data.
2. The preference data and criteria were annotated by "three PhD students in computer science" (Line 180). This is a very small and homogenous group. Their preferences, values, and interpretation of criteria are unlikely to be representative of a diverse, global user base. 3. The interpretable Omni-RewardModel-R1 is trained on only 10K samples (Line 235), which is just ~3% of the total dataset. Unsurprisingly, its performance is significantly lower than the discriminative BT model (Table 1). The paper presents this as an "exploration" but doesn't justify the decision to use such a small fraction of the data. This makes the generative model feel more like a proof-of-concept than a robust contribution.

**Strengths Contributions:**

1. The work tackles a critical and forward-looking problem. As AI models become "omni-modal," the methods to align them must follow suit.

2. The introduction of "free-form preferences" in Omni-RewardBench is a major conceptual contribution. It moves beyond simple binary choices and pushes RMs to develop a deeper, more contextual understanding of human values. The benchmark and dataset cover modalities like audio and 3D, which are often neglected in alignment research.

3. The paper evaluates an extensive set of over 30 models, providing a clear and valuable snapshot of the current capabilities and limitations of multimodal RMs. The ablation studies effectively demonstrate the importance of their contributions.

---

> ### Author Rebuttal · Authors · 2025-07-28
>
> Thank you for your careful and insightful reviews. We are deeply encouraged that you recognize the importance of aligning omni-modal AI models and view the introduction of free-form preferences as a major conceptual contribution. We sincerely hope that our response addresses your concerns.
>
> > Some limitations are listed in Appendix A. They are all valuable future directions. Besides, the work could benefit from including other important modalities like thermal, radar, tabular data, or time-series data.
>
> Thank you for your thoughtful feedback. We fully agree that incorporating additional modalities such as thermal, radar, tabular data, and time-series data would further enhance the scope and utility of our benchmark. We have included this in our future roadmap and plan to expand Omni-RewardBench to cover a broader range of modalities in subsequent versions.
>
> > The preference data and criteria were annotated by "three PhD students in computer science" (Line 180). This is a very small and homogenous group. Their preferences, values, and interpretation of criteria are unlikely to be representative of a diverse, global user base.
>
> Thank you for pointing this out. We acknowledge that the current annotation was conducted by a small group of PhD students. The annotation of the entire dataset was a time-intensive process that required substantial manual effort from our team. **Given the constraints of limited resources, we took extensive measures to ensure the quality and consistency of the annotations**. Detailed annotation guidelines and examples are provided in Appendix C.2 to ensure that all participants fully understood the process, which helped significantly mitigate potential biases.
>
> &#x20;
>
> Moreover, unlike broad and subjective preferences such as _helpfulness_ or _harmlessness_, our benchmark provides **explicit and well-defined textual criteria** for each annotation instance. **This design choice reduces the risk of ambiguity and limits the impact of cultural or individual variation in interpretation, thereby minimizing the potential issues arising from a lack of demographic diversity among annotators**.
>
> > The interpretable Omni-RewardModel-R1 is trained on only 10K samples (Line 235), which is just \~3% of the total dataset. Unsurprisingly, its performance is significantly lower than the discriminative BT model (Table 1). The paper presents this as an "exploration" but doesn't justify the decision to use such a small fraction of the data. This makes the generative model feel more like a proof-of-concept than a robust contribution.
>
> Thank you for your insightful comment. Omni-RewardModel-R1 is indeed intended as a proof-of-concept to explore the potential of using RL to elicit interpretable reward modeling capabilities in a generative model. **As noted, it was trained on 10K samples, which was primarily due to the computational cost and low sample efficiency typically associated with RL training.**
>
> Despite the limited training size, the model shows initial promise and demonstrates that reinforcement learning can elicit reward modeling capabilities in the generative model to a meaningful extent. **To further validate this direction, we expanded the training set to 20K samples in follow-up experiments. This led to a 2.5% performance improvement on Omni-RewardBench under the _w/ Tie_ setting, and a 3.2% improvement under the _w/o Tie_ setting.**
>
> In contrast, **BT models offer much higher training efficiency**, enabling them to be trained on the full dataset and thus achieve stronger performance. These results also serve as indirect evidence of the **high quality and scalability of Omni-RewardData**. This further highlights one of the key advantages of the BT framework in the context of large-scale reward modeling. While RL-based generative reward models are still in their early stages compared to discriminative BT models, we believe this remains a **valuable and necessary research direction**.

---

### Decision · Program_Chairs · 2025-09-18

**Decision:**

Reject

**Comment:**

This work proposes Omni-Reward, a new omni-modal benchmark, dataset, and model to address modality imbalance and preference rigidity in reward modeling. While the contributions are valuable and the authors clarified some of the reviewers' concerns in the rebuttal, important ethical issues, including unclear IRB approval, limited annotator representativeness, and insufficiently grounded annotation criteria, remain unresolved. Given the high acceptance bar of the NeurIPS DB track this year, the AC unfortunately recommends rejection at this stage.